# Examining Emotional Labor in COVID-19 through the Lens of Self-Efficacy

**Lixia Yao** [1] **and Jie Gao** [2,*]

1   School of Business and Administration, Zhejiang Gongshang University, Hangzhou 310018, China; yaolixia@zjgsu.edu.cn
2   Department of Hospitality, Tourism and Event Management, Lucas College and Graduate School of Business, San Jose State University, San Jose, CA 95192, USA
*   Correspondence: jie.gao@sjsu.edu

**Abstract:** The ongoing COVID-19 pandemic has dealt a significant blow to the restaurant industry, with many restaurants suspending operations or closing altogether. This study was aimed to investigate the effects of emotional labor on employees' well-being and the mediating role of self-efficacy in the context of chain restaurants. Data were collected in 2020 through an online survey in China, and results revealed that emotional labor had a significant impact on well-being that was measured in life and job satisfaction. Self-efficacy not only had a significant positive impact on employees' job- and life-related well-being but also played a fully mediating role between deep acting and life satisfaction, with a partial mediating role between deep acting and job satisfaction. Job-related well-being also played a fully mediating role between deep acting and life satisfaction, with a partial mediating role between deep acting and job satisfaction. It is important for restaurant employees to develop deep acting skills and improve self-efficacy and job satisfaction Restaurant managers must establish a healthy working environment by providing better job support and creating a more relaxed working atmosphere.

**Keywords:** COVID-19; surface acting; deep acting; self-efficacy; job satisfaction; life satisfaction

## 1. Introduction

The world has been rocked by the novel coronavirus disease 2019 (COVID-19) pandemic, which continues to wreak havoc worldwide. This ongoing global health crisis and the commensurate worldwide efforts to contain the virus' transmission by "flattening the curve" has dealt a huge blow to the restaurant industry, wherein a large number of restaurants have suspended services or even permanently closed. In 2020, the restaurant market was one of the most severely damaged industries. For example, the revenue of China's restaurant industry in 2020 was 6117 billion dollars, down 16.6% from the previous year. In China, nearly all restaurants closed their offline marketing services during this pandemic, and until now, in-store customers have comprised far fewer numbers than usual. Consequently, restaurant employees are experiencing several negative emotions, such as worries about their job security and future. Interestingly, service employees are commonly referred to as "emotional labor", because they frequently endure negative experiences with rude customers. Nonetheless, they are forced to comply with emotional expression regulations established by service-oriented organizations [1]. In this customer-oriented service environment, organizations must focus on the quality of service, particularly that of client–provider interactions. COVID-19 has greatly impacted the restaurant industry. Due to fears of losing their salaries and jobs and even being infected with COVID-19, restaurants' employees have been under much psychological pressure during the pandemic, which may be hurting their well-being. As per social cognitive theory (SCT), self-efficacy is a crucial and proximal predictor of behavior. Studies have shown that self-efficacy is an important source of happiness, and it has a strong relationship with well-being [2]. Self-efficacy has a

generative capability to organize sub-skills in a way that makes effective action possible. Therefore, beliefs about personal efficacy and not just skills are key factors in determining how a person performs in a given situation [3].

Chain restaurants emphasize brands, standardized products, and operating procedures, and franchisees are provided with assistance and benefits of technology and management so that they can offer consumers consistent products and reliable services [4]. A high chain rate indicates its brand maturity in the restaurant industry. The number of chain restaurants in China has increased over the past years [5]. However, there is limited evidence on emotional labor and employees' well-being in the context of chain restaurants are lacking. This study was thus aimed to investigate the effects of emotional labor on employees' well-being and the mediating role of self-efficiency in the context of chain restaurants. Specifically, this study had three objectives: (1) to examine two modes of emotional labor (i.e., deep acting and surface acting) and how they relate to well-being in a chain restaurant context, (2) to investigate the effects of self-efficacy on the relationship between emotional labor and well-being, and (3) to provide implications regarding how chain restaurants' management should support their employees in the demand-strained process of emotion regulation and stress coping due to the COVID-19 pandemic.

## 2. Literature Review

### 2.1. Emotional Labor

Emotional labor includes the management of emotions and expressions to comply with organizational display rules, which include two emotional labor strategies: surface acting and deep acting [6]. Many researchers have enriched and improved this topic, especially in the service industry. Two emotional labor strategies have been identified: (1) surface acting refers to employees' modification of only their observable expressions to adhere to expectations about emotional expression and (2) deep acting involves the modification of both felt and underlying emotions to adhere to rules about emotional expression [7].

Emotional labor is particularly important for interactive service jobs [8] because of the mismatch between customer expectations and staff behaviors. Judge et al. and Scott and Barnes found that employee mood was an important mechanism that could explain the relationship between emotional labor and its consequences [9,10]. Customers, especially in chain restaurants, tend to expect high-quality service from the staff, much more than surface service behaviors. Emotional labor performance has been found to be positively related to work stress [11,12]. Emotive dissonance can have negative outcomes such as low self-esteem, depression, work alienation, and burnout [13,14]. Because of its negative impact, emotional labor has been compared to "surrendering one's heart" [12]. Hence, "hypocrisy pay" for employees performing emotional labor has been advocated [15]. Employees can thus develop skills that enable both surface and deep acting [16]. Compared to surface acting, deep acting leads to a reduction in stress because the degree of emotive dissonance is forced to become lower [17]. Therefore, it is possible that training a deep acting ability can diminish the negative consequences of emotional labor on an employee's psychological health.

### 2.2. Conceptualizing the Construct of Well-Being

Well-being is a psychological state that may lead to a pleasure-filled life. The construct of well-being includes: (1) individual well-being that relates to job satisfaction and burnout and (2) organizational well-being that relates to performance and withdrawal behaviors [18]. Our study was focused on individual well-being, which reflects the level of individual psychological arousal and is thus an effective index to measure mental health.

Well-being also reflects a person's job and life satisfaction that helps maintain an effective functioning in the workplace. In this sense, well-being contributes to more than just one's state of health [19]. First, job satisfaction relies on a person's cognitive evaluation of a job's quality based on pay, coworkers, and/or supervisors. A good job, for most

individuals, can fulfill many basic needs (e.g., economic needs and relationships). Second, life satisfaction reflects an individuals' appraisal of his/her quality of life regardless of how it is achieved [20], which is a cognitive component of subjective well-being (SWB) [21]. In addition, individual motivation affects well-being, and different tasks relate to different types of well-being [22]. Individuals' pursuit of goals that are in line with their internal motives can generate higher life satisfaction.

Existing research has heavily focused on the effects of emotional labor on employees. For example, Grandey argued that emotional labor was positively related to employee burnout, dissatisfaction, and withdrawal behaviors [23]. Surface acting was found to engender emotional dissonance, an internal state of uncomfortable tension resulting from experiencing a psychological discrepancy between genuine inner feelings and feigned emotions displayed [23]. In contrast, employees who engage in deep acting consciously strive to sincerely understand their customers and empathize with them [24]. Transforming one's emotional state can involve focusing on the positive aspects of a situation, thinking about events that conjure up a desired emotion, and cognitively reappraising a situation more positively. All of these activities reflect deep-acting emotional labor [23], which could make employees increase their perception of job and life satisfaction. Employees who use a surface-acting strategy could modify their emotion expression with work pressure while their inner psychological state remained unchanged. However, employees using a deep acting strategy could make cognitive and emotional changes. The following hypotheses are thus proposed (Figure 1):

**Hypothesis 1 (H1).** *Surface acting negatively influences job satisfaction.*

**Hypothesis 2 (H2).** *Surface acting negatively influences life satisfaction.*

**Hypothesis 3 (H3).** *Deep acting positively influences job satisfaction.*

**Hypothesis 4 (H4).** *Deep acting positively influences life satisfaction.*

Employees have been evidenced to work for 8 hours per workday on average, with about 35% additional work during weekends and holidays. It is obvious that work has been filling large parts of our lives by playing a significant role, and job satisfaction thus directly influences life satisfaction. As Steve Jobs said, " . . . the only way to be truly satisfied is to do what you believe is great work". Thus, the following hypothesis is proposed (Figure 1):

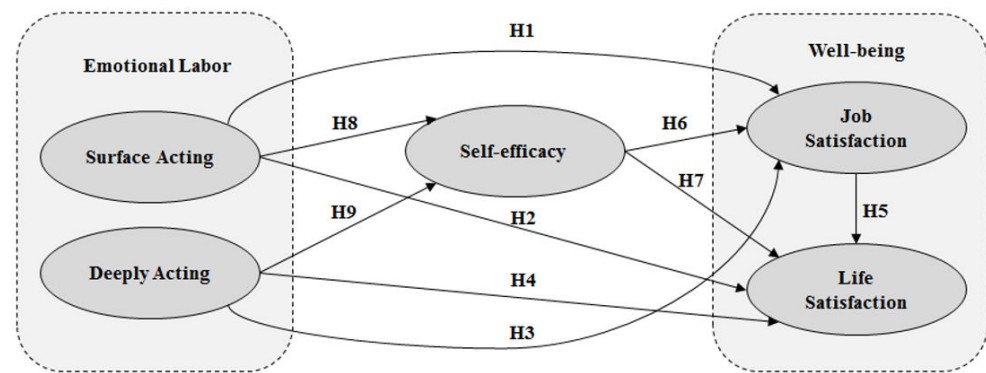

**Figure 1.** Proposed structural model. H: hypothesis.

**Hypothesis 5 (H5).** *Job satisfaction positively influences life satisfaction.*

### 2.3. Social Cognitive Theory and Self-Efficacy

SCT is one of the most common behavioral change theories used to explain human behavior. It does so in terms of a three-way, dynamic, and reciprocal model in which personal factors, environmental influences, and behaviors continually interact [25,26]. People learn through both experience and by observing the actions of others and the respective results [27].

According to Bandura, self-efficacy is "an individual's conviction (or confidence) about his or her abilities to mobilize the motivation, cognitive resources, and courses of action needed to successfully execute a specific task within a given context" [28]. SCT implies that self-efficacy is a crucial and proximal predictor of behavior. Self-efficacy beliefs indirectly affect behaviors through their impact on goal intentions. In recent years, self-efficacy has gained considerable attention in the field of organizational behavior [29]. By expanding the behavioral management approach with SCT and self-efficacy, it is expected that a more comprehensive understanding and effective management of human resources will be possible. Empirical studies have reported that self-efficacious employees are more self-confident and have positive attitudes toward their work even in difficult work situations [30,31]. Self-efficacious employees typically deliver favorable work-related performance [32,33] and are more proactive with their career choices, more effective in the decision-making process [34], and better at job attendance [35]. In contrast, employees who are not self-efficacious are more passive with their work, slackening their efforts prematurely with a higher likelihood of failure with their assigned tasks [32,36]. Meanwhile, because self-efficacious employees tend to have a more positive attitude toward their lives and work, they experience lower stress and anxiety. They are consequently less likely to be plagued by negative environmental and psychological situations. Xanthopoulou et al. reported that the negative impact of emotional demands/emotion-rule dissonance on work engagement was stronger for individuals with low self-efficacy [37].

Studies have shown that self-efficacy is an important source of happiness, and it has a strong relationship with well-being [38,39]. It differs from self-concept or self-esteem and is not meant to be a fixed ability. It can be viewed as a generative capability of organizing sub-skills in a way that makes effective action possible. Therefore, beliefs about personal efficacy and not just skills are key factors in determining how a person performs in a given situation [3]. People with low self-efficacy might believe that accidents and incidents are more complicated and dire than they really are. This aspect can lead to anxiety, depression, helplessness, reduced academic performance, or low motivation. Rodebaugh and Goldin et al. showed that in socially anxious individuals, lower self-efficacy predicts poorer behavioral performance of speech tasks and interactions with others [40,41]. However, high self-efficacy is related, for example, to the regulation of stress, higher self-esteem, better well-being, good physical conditioning, and adaptation to and recovery from acute and chronic diseases [42,43]. Moreover, a high sense of self-efficacy is associated with positive feelings about one's self, which facilitate cognitive processes, academic achievements, confidence, and motivation [44]. Raggi et al. revealed that having a sense of control over events and high self-efficacy leads to higher levels of psychological well-being and life satisfaction [45]. People possessing feelings of high control over their lives and those who have achieved self-recognition have greater confidence in their abilities, better health, good well-being, and high satisfaction [39].

In summary, self-efficacy can effectively predict SWB. Salami's finding represented a significant relationship between self-efficacy, happiness, and well-being [46].

Based on the above theoretical analysis, the authors of this study posit the following hypotheses (see Figure 1):

**Hypothesis 6 (H6).** *There is a positive correlation between self-efficacy and job satisfaction.*

**Hypothesis 7 (H7).** *There is a positive correlation between self-efficacy and life satisfaction.*

## 2.4. Self-Efficacy and Its Mediating Role

According to Hochschild, emotional labor can cause feelings of estrangement, alienation, and inauthenticity due to the gap between felt and feigned emotions [6]. Thus, stress is likely to emerge. According to SCT, self-efficacy is built on the experience of self-mastery, implying that personal successes can instill perceptions of self-efficacy [31]. Following this logic, a good understanding of display rules and successful emotive acting can help employees see themselves as efficacious on the job because service interactions become more predictable and unpleasant service encounters can be averted [47].

Although individuals can perform under unfavorable circumstances due to self-mastery, it is less likely for those employees to create and maintain a positive psychological status toward the self [48]. According to the job demand–resources model of occupational stress, which explains how job resources decrease the likelihood of employee disengagement even after employees encounter overload [49] when perceiving negative display rules and surface acting are embodied in work roles, the provision of job resources can subdue negative reactions to self-estrangement, alienation, and inauthenticity. Self-efficacy is thus deemed an important personal resource with which one can cope with job stress. Therefore, an increase in perceived self-efficacy may enable employees to view emotional labor as a means to an end (i.e., rewards and recognition) rather than a stressful act [50,51]. Thus, the following hypotheses are proposed (Figure 1):

**Hypothesis 8 (H8).** *Surface acting has a negative correlation with self-efficacy.*

**Hypothesis 9 (H9).** *Deep acting has a positive correlation with self-efficacy.*

Self-efficacy affects the relationship between emotional labor and employees' well-being [50,52]. Emotional labor requires employees to suppress their inner feelings in order to display appropriate emotions, which may make employees feel inauthentic. However, employees may offset the costs of false emotional displays in a goal of achieving a successful emotional management [50,52]. According to Hsieh and Guy, emotional labor is a double-edged sword [53], and self-efficacy could be the mechanism used to help employees effectively wield it [47]. Hsieh examined the role of job resources in explaining the effects of emotional labor requirements on burnout and found that social support successfully mediates the deleterious effect of negative display rules [54]. Self-efficacy is an important personal resource to deal with the drawbacks of emotional labor. Therefore, additional hypotheses are proposed:

**Hypothesis 9a (H9a).** *Self-efficacy plays a mediating role in the relationship between surface acting and job satisfaction.*

**Hypothesis 9b (H9b).** *Self-efficacy plays a mediating role in the relationship between surface acting and life satisfaction.*

**Hypothesis 9c (H9c).** *Self-efficacy plays a mediating role in the relationship between deep acting and job satisfaction.*

**Hypothesis 9d (H9d).** *Self-efficacy plays a mediating role in the relationship between deep acting and life satisfaction.*

## 3. Research Methods

### 3.1. Data Collection

Based on the study's aim and objectives, an electronic survey was developed to collect data on (1) emotional labor, the instrument of which was developed by Gandey and Dfendorff [55]; (2) self-efficacy, the instrument of which was Schwarzer and Jerusalem's Perceived Self-Efficacy Scale [56]; (3) job satisfaction, the instrument of which was developed

based on Sharma et al.'s work [57]; (4) life satisfaction, the instrument of which was built by Diener et al. [58]; and (5) participants' socio-demographic profile, including gender, age, years of work experience, marital status, working category, and education level (Table 1). A 5-point Likert-type response scale was used for these items (from 1 = strongly disagree to 5 = strongly agree). It is worth noting that emotional labor has been evidenced to influence employees' well-being [23,24,44]. Self-efficacy may affect the relationship between emotional labor and worker well-being [59,60]. For the job satisfaction instrument, we decided to choose four items, i.e., pay, training, promotion, and recognition, based on Chinese collectivist culture, and we eliminated the other two (i.e., supervision and job security).

**Table 1.** Profile of the respondents.

| | *N* | % |
|---|---|---|
| **Gender (*n* = 331)** | | |
| Male | 175 | 52.87 |
| Female | 156 | 47.13 |
| **Age (*n* = 331)** | | |
| Under 20 | 10 | 3.02 |
| 20–30 | 170 | 51.36 |
| 31–40 | 119 | 35.95 |
| 41–50 | 26 | 7.85 |
| Over 50 | 6 | 1.81 |
| **Highest education (*n* = 331)** | | |
| Less than high school degree | 9 | 2.72 |
| High school degree | 37 | 11.18 |
| College degree | 268 | 80.97 |
| Master's degree or above | 17 | 5.14 |
| **Working Category (*n* = 331)** | | |
| Front line employees | 143 | 43.20 |
| Supervisor | 69 | 20.85 |
| Manager | 69 | 20.85 |
| Logistics | 50 | 15.11 |
| **Years of durations in the position (*n* = 331)** | | |
| Less than one year | 76 | 22.96 |
| 1–3 years | 125 | 37.76 |
| 3–5 years | 63 | 19.03 |
| More than 5 years | 67 | 20.24 |
| **Marital status (*n* = 331)** | | |
| Married with children | 134 | 40.48 |
| Married and no child | 34 | 10.27 |
| Single with child or children | 7 | 2.11 |
| Single and no child | 156 | 47.13 |
| **Annual Income (*n* = 331)** | | |
| Less than 30,000 | 82 | 24.77 |
| 30,000–50,000 | 91 | 27.49 |
| 50,000–100,000 | 103 | 31.12 |
| 100,000–200,000 | 45 | 13.60 |
| More than 200,000 | 10 | 3.02 |

The online survey was launched in March and April 2020. We purchased the sample from a contracted marketing agency, who emailed the survey link to their participants. The participants in our survey had to be current employees working for Chinese chain restaurants of diverse age and gender. When participants clicked on the survey link, they first accessed a page that described the study purpose, confidentiality, and privacy protocols. During our contract with the marketing agency, we collected a total of 335

questionnaires, four of which were eliminated due to incomplete responses. The process led to our final sample size of 331.

*3.2. Data Analysis*

The data analysis in this study was divided into five steps. First, descriptive statistics were analyzed to determine the characteristics and distribution of the measured variables and the reliability of the scale was examined. Second, the confirmatory factor analysis statistics were applied to test the goodness of fit about the items and assumptions and the reliability of the scale was examined. Third, correlations between variables were analyzed and the significance was tested. Fourth, a measurement model with all constructs/dimensions was established using confirmatory factor analysis (CFA) in an attempt to test the fit of the measures. In the next step, a baseline path model was developed using structural equation modeling (SEM) to test H1–H9. SEM was used to test the relationship between the independent and dependent variables. For mediation to occur, four criteria had to be met: (1) the independent variable should be significantly associated with the dependent variable, (2) the independent variable should be related to the mediator, (3) the mediator should be related to the dependent variable, and (4) the association between the independent and dependent variables must be reduced when the mediator is partially omitted [61]. Here, all variables (surface acting, deep acting, self-efficacy, job satisfaction, and life satisfaction) were included in the measurement model (as shown in Figure 1), and the goodness of fit was tested with SEM to explore the path of influence and the extent to which the dependent variable could be explained. Finally, a structural model was developed, and the regression weights were compared to test the hypotheses.

## 4. Results

*4.1. Respondents' Sociodemographic Profiles*

Among the respondents, the percentage of men (52.87%) and women (47.13%) was approximately similar (Table 1), while the mean age and years of work experience were 32.4 years (SD = 9.8) and 4.0 years (SD = 3.3), respectively. In fact, more than half of the respondents were between 20 and 30 years old (51.36%), followed by those between 30 and 40 years old (35.95%). Regarding their occupations, the majority were frontline employees (43.20%), followed by supervisors (20.85%), managers (20.85%), and logistics employees (15.11%).

Most of the respondents (80.97%) had a bachelor's degree. Approximately half (47.13%) were single with no children, and 40.48% were married with children. In terms of work experience, more than half of the respondents (60.72%) had less than 3 years of experience and approximately one-third (33.7%) had 4 to 6 years of work experience.

*4.2. Descriptive Statistics and Correlations for the Measured Variables*

The Cronbach's $\alpha$ tests showed a high reliability for the measures of surface acting ($\alpha = 0.80$) and deep acting ($\alpha = 0.84$), which were higher than the suggested value of 0.80 [61]. Bartlett's Test of Sphericity for the self-efficacy scale presented a value of 417.64, with a significance level below 0.001; the Kaiser–Meyer–Olkin (KMO) values for the analyses were 0.82, which was higher than 0.80; and Bartlett's Test of Sphericity was significant at the 0.001 level. The results of the reliability analysis indicated high internal consistency for destination quality. The AVE (average variance extracted) values for all four factors were 0.63 and 0.66, which were higher than the suggested value of 0.50 [62]. Thus, the convergent validity of the scale was acceptable. The respondents were likely to agree with the following statements about EL: "I work hard to feel the emotions that I need to show to customers." (M = 3.83); "I make an effort to actually feel the emotions that I need to display toward others." (M = 3.75); "I try to actually experience the emotions that I show to customers." (M = 3.74); and "I work at developing the feelings inside of me that I need to show to customers." (M = 3.70). All of the items for EL were above 3.0 (on average), which showed that the restaurant employees were making such efforts in their customer service roles (Table 2).

**Table 2.** Descriptive statistics about the respondents' use of emotional labor strategies and perceived well-being and self-efficacy.

| Items | M | SD |
| --- | --- | --- |
| **Surface Acting (α = 0.80)** | | |
| 1. I put on an act in order to deal with customers in an appropriate way. | 3.54 | 1.00 |
| 2. I fake a good mood when interacting with customers. | 3.62 | 1.02 |
| 3. I put on a "show" or "performance" when interacting with customers. | 2.92 | 1.15 |
| 4. I just pretend to have the emotions I need to display for my job. | 2.89 | 1.10 |
| 5. I put on a "mask" in order to display the emotions I need for the job. | 3.32 | 1.16 |
| 6. I show feelings to customers that are different from what I feel inside. | 3.14 | 1.06 |
| 7. I fake the emotions I show when dealing with customers. | 2.68 | 1.11 |
| **Deep Acting (α = 0.84)** | | |
| 8. I try to actually experience the emotions that I must show to customers. | 3.74 | 0.89 |
| 9. I make an effort to actually feel the emotions that I need to display toward others. | 3.75 | 0.89 |
| 10. I work hard to feel the emotions that I need to show to customers. | 3.83 | 0.93 |
| 11. I work at developing the feelings inside of me that I need to show to customers. | 3.70 | 0.96 |
| **Self-Efficacy (α = 0.84)** | | |
| 1. I can always manage to solve difficult problems if I try hard enough. | 3.87 | 0.98 |
| 2. If someone opposes me, I can find the means and ways to get what I want. | 3.81 | 0.90 |
| 3. It is easy for me to stick to my aims and accomplish my goals. | 3.60 | 1.01 |
| 4. I am confident that I could deal efficiently with unexpected events. | 3.77 | 0.94 |
| 5. Thanks to my resourcefulness, I know how to handle unforeseen situations. | 3.50 | 0.95 |
| 6. I can solve most problems if I invest the necessary effort. | 3.87 | 0.92 |
| 7. I can remain calm when facing difficulties because I can rely on my coping abilities. | 3.83 | 0.87 |
| 8. When I am confronted with a problem, I can usually find several solutions. | 3.73 | 0.92 |
| 9. If I am in trouble, I can usually think of a solution. | 3.80 | 0.86 |
| 10. I can usually handle whatever comes my way. | 3.13 | 1.05 |
| **Job Satisfaction (α = 0.90)** | | |
| 1. I feel I am being paid a fair amount for the work I do. | 3.34 | 1.02 |
| 2. Raises are too far and few between. | 2.16 | 0.97 |
| 3. I am unappreciated by the organization when I think about what they pay me. | 2.71 | 0.97 |
| 4. I feel satisfied with my chance for salary increases | 3.23 | 1.16 |
| 5. The company only gives people the minimum amount of training they need to do their job. | 2.66 | 1.05 |
| 6. People are strongly encouraged to develop their skills. | 3.92 | 0.92 |
| 7. Adequate on-the-job training was provided to internal user groups to use the new system. | 3.64 | 0.98 |
| 8. Both technology and process training were provided to employees using the system. | 3.80 | 0.93 |
| 9. There is really too little chance for promotion on my job. | 2.44 | 1.00 |
| 10. Those who do well on the job stand a fair chance of being promoted. | 3.79 | 1.06 |
| 11. I am satisfied with my chances for promotion. | 3.11 | 1.09 |
| 12. People get ahead as fast here as they do in other places. | 3.18 | 1.06 |
| 13. I am satisfied with the recognition for my work from superiors. | 3.59 | 0.98 |
| 14. I am happy with the amount of encouragement and positive feedback. | 2.98 | 1.08 |
| **Life Satisfaction (α = 0.83)** | | |
| 15. The conditions of my life are excellent. | 2.71 | 1.10 |
| 16. I am satisfied with my life. | 3.01 | 1.08 |
| 17. So far I have gotten the important things I want in life. | 2.79 | 1.15 |
| 18. If I could live my life over, I would change almost nothing. | 3.84 | 1.22 |

Additionally, the measure of self-efficacy had a high reliability (α = 0.84). Bartlett's Test of Sphericity for the self-efficacy scale presented a value of 849.75, with a significance level below 0.005, and the KMO value was 0.89, which showed high validity (Table 3). The respondents were likely to agree with the following items: "I can always manage to solve difficult problems if I try hard enough." (M = 3.87); "I can solve most problems if I invest the necessary effort." (M = 3.87); "I can remain calm when facing difficulties because I can rely on my coping abilities." (M = 3.83); "If someone opposes me, then I can find the means and ways to get what I want." (M = 3.81); and "If I am in trouble, then I can usually think of a solution." (M = 3.80). Moreover, the result of Bartlett's Test of Sphericity for the well-being scale was 1795.63 (171), with a significance below 0.001, and the KMO value was 0.87, indicating high validity (Table 3).

**Table 3.** Results of the confirmatory factor analysis.

| Item | Factor Loading | Error Variances | CR | Ave | Cronbach's $\alpha$ | Bartlett Sphericity Test (Df) | KMO |
|---|---|---|---|---|---|---|---|
| **Surface Acting** | | | | | | | |
| 1. I put on an act in order to deal with customers in an appropriate way. | 0.64 | 0.81 | | | | | |
| 2. I fake a good mood when interacting with customers. | 0.80 | 0.62 | | | | | |
| 3. I put on a "show" or "performance" when interacting with customers. | 0.79 | 0.63 | 0.78 | 0.63 | 0.80 | | |
| 4. I just pretend to have the emotions I need to display for my job. | 0.77 | 0.66 | | | | | |
| 5. I put on a "mask" in order to display the emotions I need for the job. | 0.74 | 0.70 | | | | 417.637(10) *** | 0.82 |
| 6. I show feelings to customers that are different from what I feel inside. | 0.76 | 0.70 | | | | | |
| 7. I fake the emotions I show when dealing with customers. | 0.83 | 0.69 | | | | | |
| **Deep Acting** | | | | | | | |
| 8. I try to actually experience the emotions that I must show to customers. | 0.76 | 0.57 | | | | | |
| 9. I make an effort to actually feel the emotions that I need to display toward others. | 0.83 | 0.47 | 0.78 | 0.66 | 0.84 | | |
| 10. I work hard to feel the emotions that I need to show to customers. | 0.82 | 0.49 | | | | | |
| 11. I work at developing the feelings inside of me that I need to show to customers. | 0.72 | 0.62 | | | | | |
| **Self-Efficacy** | | | | | | | |
| 1. I can always manage to solve difficult problems if I try hard enough. | 0.61 | 0.74 | | | | | |
| 2. If someone opposes me, I can find the means and ways to get what I want. | 0.65 | 0.70 | | | | | |
| 3. It is easy for me to stick to my aims and accomplish my goals. | 0.79 | 0.53 | | | | | |
| 4. I am confident that I could deal efficiently with unexpected events. | 0.73 | 0.60 | | | | | |
| 5. Thanks to my resourcefulness, I know how to handle unforeseen situations. | 0.78 | 0.67 | 0.83 | 0.55 | 0.84 | 849.75(45) ** | 0.892 |
| 6. I can solve most problems if I invest the necessary effort. | 0.70 | 0.64 | | | | | |
| 7. I can remain calm when facing difficulties because I can rely on my coping abilities. | 0.65 | 0.70 | | | | | |
| 8. When I am confronted with a problem, I can usually find several solutions. | 0.72 | 0.61 | | | | | |
| 9. If I am in trouble, I can usually think of a solution. | 0.65 | 0.69 | | | | | |
| 10. I can usually handle whatever comes my way. | 0.61 | 0.74 | | | | | |
| **Job Satisfaction** | | | | | | | |
| 1. I feel I am being paid a fair amount for the work I do. | 0.55 | 0.43 | | | | | |
| 2. Raises are too far and few between. | 0.81 | 0.57 | | | | | |
| 3. I am unappreciated by the organization when I think about what they pay me. | 0.74 | 0.46 | | | | | |
| 4. I feel satisfied with my chance for salary increases | 0.66 | 0.54 | | | | | |
| 5. The company only gives people the minimum amount of training they need to do their job. | 0.68 | 0.59 | | | | | |
| 6. People are strongly encouraged to develop their skills. | 0.68 | 0.37 | | | | | |
| 7. Adequate on-the-job training was provided to internal user groups to use the new system. | 0.72 | 0.49 | 0.60 | 0.57 | 0.90 | | |
| 8. Both technology and process training were provided to employees using the system. | 0.76 | 0.52 | | | | | |
| 9. There is really too little chance for promotion on my job. | 0.89 | 0.63 | | | | | |
| 10. Those who do well on the job stand a fair chance of being promoted. | 0.84 | 0.53 | | | | 1795.63(171) ** | 0.87 |
| 11. I am satisfied with my chances for promotion. | 0.84 | 0.59 | | | | | |
| 12. People get ahead as fast here as they do in other places. | 0.73 | 0.51 | | | | | |
| 13. I am satisfied with the recognition for my work from superiors. | 0.79 | 0.52 | | | | | |
| 14. I am happy with the amount of encouragement and positive feedback. | 0.82 | 0.42 | | | | | |
| **Life Satisfaction** | | | | | | | |
| 15. The conditions of my life are excellent. | 0.76 | 0.54 | | | | | |
| 16. I am satisfied with my life. | 0.72 | 0.60 | 0.63 | 0.58 | 0.83 | | |
| 17. So far I have gotten the important things I want in life. | 0.76 | 0.57 | | | | | |
| 18. If I could live my life over, I would change almost nothing. | 0.70 | 0.58 | | | | | |

Notes: All factor loadings were significant at the level of 0.001 or above. ** $p < 0.01$, and *** $p < 0.001$. Model fit indices: $n = 331$; $\chi^2 = 536.295$; df = 289; CFI = 0.900; GFI = 0.886; NFI = 0.809; and RMSEA = 0.051. AVE: average variance extracted.

The Cronbach's $\alpha$ tests also showed a high reliability for the measure of job satisfaction ($\alpha$= 0.90) and life satisfaction ($\alpha$ = 0.83). Bartlett's Test of Sphericity for the well-being scale presented a value of 1795.63, with a significance level below 0.05, and the KMO value was 0.87, which showed high validity (Table 3). The respondents were likely to agree with the following items: "People are strongly encouraged to develop their skills." (M = 3.92); "If I could live my life over, then I would change almost nothing." (M = 3.84); and "Both technology and process training were provided to employees using the system." (M = 3.80). Meanwhile, the respondents generally disagreed with the following items: "Raises are too far and few between." (M = 2.16); and "There are few opportunities for promotions in my job." (M = 2.44) (Table 2).

The Cronbach's $\alpha$ value for each variable was greater than 0.80, which indicated that the variables had a high internal consistency and that the questionnaire had good reliability (Table 4). According to the significance level and KMO metrics, the correlation coefficient matrix and the identity matrix of all variables were significantly different, and the original variable-factor analysis was appropriate.

**Table 4.** Correlations between variables.

|  | Surface Acting | Deep Acting | Self-Efficacy | Job Satisfaction | Life Satisfaction |
|---|---|---|---|---|---|
| **Surface Acting** | 1.00 |  |  |  |  |
| **Deep Acting** | 0.01 | 1.00 |  |  |  |
| **Self-Efficacy** | −0.11 * | 0.34 ** | 1.00 |  |  |
| **Job Satisfaction** | −0.20 ** | 0.20 ** | 0.33 ** | 1.00 |  |
| **Life Satisfaction** | −0.09 | 0.15 ** | 0.40 ** | 0.44 ** | 1.00 |

Note: All factor loadings were significant at the level of 0.01 or above. * $p < 0.05$ and ** $p < 0.01$.

We analyzed the correlations between the five variables and found the following relationships: surface acting and job satisfaction had a significant negative correlation (r = −0.20; $p < 0.01$), surface acting and self-efficacy had a significant negative correlation (r = −0.11; $p < 0.1$), deep acting and self-efficacy had a significant positive correlation (r = 0.34; $p < 0.01$), deep acting and job satisfaction had a significant positive correlation (r = 0.20; $p < 0.01$), deep acting and life satisfaction had a significant positive correlation (r = 0.15; $p < 0.01$), self-efficacy and job satisfaction had a significant positive correlation (r = 0.33; $p < 0.01$), self-efficacy and life satisfaction had a significant positive correlation (r = 0.40; $p < 0.01$), and job satisfaction and life satisfaction had a significant positive correlation (r = 0.44; $p < 0.01$) (see Table 4). We established a baseline structural equation model, shown below, to explore their relations in depth.

*4.3. Measurement Model*

In the second step, the fits of the measures were assessed using a CFA. In this research, model fits were evaluated by means of the comparative fit index (CFI), the goodness-of-fit index (GFI), the normed fit index (NFI), and the root mean square error of approximation (RMSEA). Based on Byrne's and Bollen's suggestions, a model is regarded as acceptable if the CFI exceeds 0.93, if the NFI and the GFI exceed 0.90, and if the RMSEA is less than 0.80 [62,63]. Thus, the results of the confirmatory factor analysis in the present study revealed that the initial measurement model, consisting of 11 items for two factors (surface acting and deep acting), 9 items for one factor (self-efficacy), and 19 items for two factors (job satisfaction and life satisfaction), had acceptable fit indices ($n$ = 331; $\chi^2$ = 536.295; df = 289; CFI = 0.900; GFI = 0.886; NFI = 0.809; and RMSEA = 0.051).

Furthermore, all regression weights were significant ($p < 0.001$). Thus, the measurement model was further refined, as standardized residuals greater than 2.57 and large modification indices greater than 3.84 are considered statistically significant [63]. The discriminant validity of the measures was also assessed by comparing the squares of the correlations between each pair of factors with their AVEs [62]. Meanwhile, the correlation coefficients between the five factors ranged from 0.28 to 0.62. As the AVEs for all

seven latent factors were higher than the squares of all of the correlation coefficients, the discriminant validity was considered to be acceptable. As shown in Tables 4 and 5, the composite reliability values for surface acting, deep acting, self-efficacy, job satisfaction, and life satisfaction were 0.80, 0.84, 0.84, and 0.90, respectively, indicating that all of the values were higher than the suggested 0.80 value [63]. Moreover, the AVEs for all seven factors were equal to or higher than the suggested 0.50 value [62], indicating that the convergent validity of the scale was acceptable.

**Table 5.** Results of SEM.

| | Standardized Direct Effects | Standardized Indirect Effects | Standardized Total Direct Effects | Standardized Indirect Effects (95%CI) | S.E. | C.R. |
|---|---|---|---|---|---|---|
| H1: Surface Acting→Job Satisfaction | −0.146 * | −0.054 | −0.146 | | 0.063 | −2.285 |
| H2: Surface Acting→Life Satisfaction | −0.042 | −0.063 | −0.044 | | 0.067 | −0.652 |
| H3: Deep Acting→Job Satisfaction | 0.226 ** | 0.168 | 0.226 | 0.168 (0.098–0.294) ** | 0.089 | 3.069 |
| H4: Deep Acting→Life Satisfaction | −0.078 | 0.185 | −0.078 | 0.185 (0.088–0.313) ** | 0.095 | −0.993 |
| H5: Job Satisfaction→Life Satisfaction | 0.054 | 0 | 0.054 | | 0.089 | 0.613 |
| H6: Self-Efficacy→Job Satisfaction | 0.419 *** | 0 | 0.419 | | 0.1 | 5.13 |
| H7: Self-Efficacy→Life Satisfaction | 0.408 *** | 0.023 | 0.408 | | 0.118 | 4.259 |
| H8: Surface Acting→Self-Efficacy | −0.128 | 0 | −0.128 | | 0.053 | −1.939 |
| H9: Deep Acting→Self-Efficacy | 0.402 *** | 0 | 0.402 | | 0.079 | 5.001 |

Notes: 1 All factor loadings were significant at the level of 0.001 or above. * $p < 0.05$, ** $p < 0.01$, and *** $p < 0.001$. 2 Model fit indices: $n = 331$; $\chi^2 = 536.295$; df = 289; CFI = 0.900; GFI = 0.886; NFI = 0.809; and RMSEA = 0.051. AVE: average variance extracted.

### 4.4. Baseline Model

In the next step, H1–H9 were tested by establishing a baseline structural equation model ($n = 331$) that included EL as an exogenous variable and self-efficacy, job satisfaction, and life satisfaction as endogenous variables. To simplify the data analysis, the factor scores for the five factors (surface acting, deep acting, self-efficacy, job satisfaction, and life satisfaction) were used in the structural equation model. As shown in Table 5 and Figure 2, the baseline model had acceptable fit indices ($n = 331$; $\chi^2 = 536.295$; df = 289; CFI = 0.900; GFI = 0.886; NFI = 0.809; and RMSEA = 0.051). The results also revealed that surface acting had a significant negative direct effect on job satisfaction ($\beta = -0.146$; $p < 0.1$). Thus, H1 is supported. In addition, as deep acting had a significant direct effect on job satisfaction ($\beta = 0.226$; $p < 0.01$) and self-efficacy ($\beta = 0.402$; $p < 0.001$), H3 and H9 are supported. Self-efficacy had a significant direct effect on job satisfaction ($\beta = 0.419$; $p < 0.001$) and life satisfaction ($\beta = 0.408$; $p < 0.001$). Hence, H6 and H7 are supported.

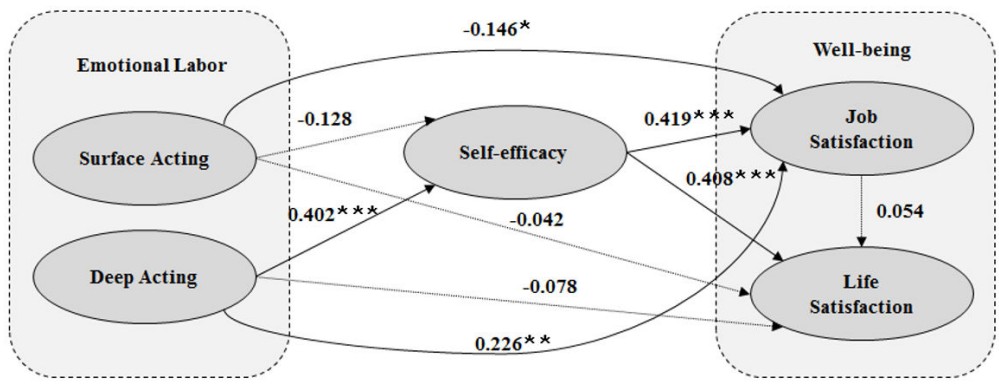

**Figure 2.** Results of structural modeling. Notes: 1 All factor loadings were significant at the level of 0.001 or above. * $p < 0.05$, ** $p < 0.01$, and *** $p < 0.001$. 2 Model fit indices: $n = 331$; $\chi^2 = 536.295$; df = 289; CFI = 0.900; GFI = 0.886; NFI = 0.809; and RMSEA = 0.051. AVE: average variance extracted.

The results also showed that surface acting had no significant direct effect on self-efficacy and life satisfaction, so H8and H2 are not supported. Deep acting had no significant direct effect on life satisfaction, so H4 is not supported.

Furthermore, we used the bootstrap resampling technique to examine the mediating role of self-efficacy. In recent years, most papers published in top academic journals in the fields of psychology, consumer behavior, and organizational behavior have used the bootstrap method to test the mediating effect. The bootstrap method first assumes that the samples obtained in a study can represent the population. Then, the samples obtained are regarded as the population of the bootstrapping, and repeated sampling is conducted from the population to generate many bootstrap samples (such as 5000 times). The 95% confidence interval of the coefficient product in the correlation equation was calculated based on the coefficient estimation of bootstrap samples. If the confidence interval did not include 0, the mediation effect was considered to be significant [64]. Therefore, we first used surface acting and deep acting as independent variables and job satisfaction as the dependent variable, with self-efficacy as the mediating variable. Second, according to bootstrap method, the sample size was selected as 5000 and a 95% bootstrap confidence interval was applied to examine the mediating effect of self-efficacy. The results exhibited that the coefficients of deep acting and job satisfaction did not contain 0, and the mediating effect was significant ($\beta = 0.17$; SE = 0.089; 95% CI = 0.098–0.294; $p < 0.05$). Deep acting also had a significant direct effect on job satisfaction ($\beta = 0.226$; $p < 0.01$) and self-efficacy ($\beta = 0.402$; $p < 0.001$), while self-efficacy had a significant direct effect on job satisfaction ($\beta = 0.419$; $p < 0.001$), with a partial mediating role between them. Similarly, the coefficients of deep acting and life satisfaction did not contain 0, and the mediating effect was significant ($\beta = 0.185$; SE = 0.095; 95% CI = 0.088–0.313; $p < 0.05$). Deep acting had a significant direct effect on job satisfaction ($\beta = 0.226$; $p < 0.01$) and self-efficacy ($\beta = 0.402$; $p < 0.001$), while self-efficacy had a significant direct effect on job satisfaction ($\beta = 0.419$; $p < 0.001$) and life satisfaction ($\beta = 0.408$; $p < 0.001$). However, deep acting and job satisfaction had no significant direct effect on life satisfaction. Thus, self-efficacy and job satisfaction were found to play a fully mediating role between deep acting and life satisfaction.

## 5. Discussion and Conclusions

The authors of this study investigated the relationship between emotional labor and well-being among chain restaurant staff who have been hit hardest by the COVID-19 pandemic and explored the mechanisms of job- and life-related well-being. The authors further examined the mediating effects of self-efficacy and job satisfaction.

The study findings provide the following theoretical implications. First, most previous studies discussed the relationship between EL and employee well-being, whereas this study explored the mechanism of the mediating role of self-efficacy, thus representing a useful addition for the relevant theoretical research. According to SCT, self-efficacy is a subjective judgment by an individual that is crucial to affecting their self-regulation to influence which challenges they decide to meet and how high they set their goals. Our study results also support this proposition. Self-efficacy has a key positive impact on well-being and plays a mediating role in the relationship between emotional labor and well-being. Thus, the promotion of employees' self-efficacy is the key to improving their emotional labor and well-being. Schunk et al. demonstrated that vicarious experiences (e.g., modeling) and verbal persuasion were two sources from which people could gain efficacy information [65]. Managers can either verbally encourage their employees to conduct this task or they can establish role models to help them increase their confidence at work [59]. Barriers to the development of self-efficacy include repeated failures to perform and meet demands, poor feedback, and faulty organization. In the context of COVID-19, the restaurant industry has suffered considerably. Thus, it is vital to provide measures to help employees manage their anxiety and facilitate psychological support. Because self-efficacy had a large positive effect on psychological well-being ($p < 0.001$), organizations require high-quality verbal

validation and feedback, positive placement experiences, positive coping styles, and strong internal loci of control to increase self-efficacy.

Second, this study has clarified the underlying mechanism of the two types of well-being and how they are affected by employees' emotional labor and self-efficacy. Erdogan et al. asked "What is the contribution of the work domain to life satisfaction?" and then concluded "Interestingly, the field of management does not have a ready answer to this important question [66]". There has thus been a notable absence in the research regarding an inner mechanism that influences job and life satisfaction. These results have evidenced the importance of job satisfaction, which plays a fully mediating role in the relationship between deep acting and life satisfaction. This information is thus quite valuable for the restaurant industry and perhaps all industries. COVID-19 has caused great social stress, thus causing people's well-being to sharply decline because of the fears related to economic downturn and disease. Restoring confidence in the future is the key to boosting the economy. This study revealed how we can improve people's life satisfaction by promoting greater well-being attained by improving their job satisfaction. Thus, entrepreneurs and policy makers should aim to create good working environments with a more relaxed working atmosphere, better job support, and the smallest unemployment rate possible.

Third, this study has clarified the underlying mechanism regarding the two types of emotional labor and their different effects on employee well-being. Although previous studies on emotional labor have demonstrated that deep acting was much more valuable than surface acting, the authors of this study explored the underlying mechanism regarding the mediating role of self-efficacy in this relationship [11,67,68]. The study results showed that deep acting has a positive effect on self-efficacy and job satisfaction. However, surface acting was found to have a negative effect on job satisfaction. Regarding SCT, self-efficacy was found to be an important resource for promoting resilience [69]. These study findings have practical implications for service industries, and they imply that deep acting might be more efficient than surface acting in aiding service employees' well-being. Therefore, training employees about deep acting is extremely important regarding its long-term impact on employees' well-being.

Finally, the authors of this study examined the mechanism of emotional labor on well-being under special circumstances. The ongoing global health crisis has caused unprecedented challenges for all employees. This study has revealed that as an effective regulation of emotions, deep acting is crucial to enhancing the well-being of both work and life. Previous research has emphasized the importance of actively employing emotional appraisal to cope with difficult situations [70]. There may be great deal of pressure regarding work and family being imposed on employees. Thus, reducing the unemployment rate as much as possible, improving employees' confidence in the future, and conducting deep acting training are all effective methods to improve people's well-being and help society escape from the economic crisis.

Given the enormous shock that the pandemic has brought to employees and the industry, many restaurant employees have been experiencing much psychological stress. Restaurant managers should actively support their employees by helping them achieve a positive psychological state of mind and learned optimism. These study results provide effective strategies to restaurant managers regarding how to support their employees, improve their well-being, and live through the hard times together.

This study was aimed to investigate the effects of emotional labor on employees' well-being and the mediating role of self-efficacy in the context of chain restaurants. Given the cultural differences in China and Western countries, this study's results should be applied with caution in other cultural contexts. Future research should account for the factor of culture and examine the role of culture in employees' emotional management and well-being. In addition, future researchers may want to study organizational cultural and situational factors, as well as their influences on job satisfaction and life satisfaction, particularly in the pandemic context. Lastly, future research should examine these constructs in

other contexts, such as hotels, resorts, and airline companies, and see how their employees manage emotions and well-being in response to work stress in addition to the pandemic.

**Author Contributions:** Conceptualization, J.G.; methodology, J.G. and L.Y.; data collection, L.Y.; formal analysis, L.Y.; manuscript writing, L.Y and J.G..; visualization, J.G. All authors have read and agreed to the published version of the manuscript.

**Funding:** This research was funded by the project (20Y092) funded by philosophy and social science research from the Education Department of Hubei Province.

**Informed Consent Statement:** Informed consent was obtained from all subjects involved in the study.

**Conflicts of Interest:** The authors declare no conflict of interest.

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
