# Peer review of "Examining Emotional Labor in COVID-19 through the Lens of Self-Efficacy"

_sustainability, doi:10.3390/su132413674_

Round 1

Reviewer 1 Report

The references are not presented in alphabetical order.

I think the results are independent from the Covid-19 pandemic and crisis. I think that findings just could be stressed for the pandemic, but not changed. I would not do so much reference to the pandemic crisis because it is not so relevant for the study and its findings.

It is interesting for the implementation of measures to prevent psychosocial risks in that sector.

Author Response

Point 1: I think the results are independent from the Covid-19 pandemic and crisis. I think that findings just could be stressed for the pandemic, but not changed. I would not do so much reference to the pandemic crisis because it is not so relevant for the study and its findings.

Response 1: Thank you for the comments. We have revised our results section to be clearer. Since our data were collected during the pandemic, our results were discussed in the context of the COVID pandemic, but were not directly related to COVID. We also deleted some unrelated references to COVID.

Please see our explanation on Page 28 and the references on Pages 29-36. (in red).

Point 2: The references are not presented in alphabetical order.

Response 2: We have fixed the references in the alphabetical order.

 Please see the references section. (in red).

Reviewer 2 Report

Dear Authors

The article deals with an important and actual topic. I would like to present some opinions. I hope they will be useful for you.

  1. The difficulty in reading and evaluating the article is the fact that the tables and figures referenced in the text are not included. Perhaps the correct version of the text was not submitted for review. The above problem is very evident in the methods and results section of the article.
  2. A lot of detailed hypotheses were formulated in the article. It would be easier for the reader to visualize them graphically. The missing figure 1 certainly contains valuable information on the measurement model. It seems that not all the hypotheses have sufficiently strong roots in the literature.
  3. I believe that the part of work devoted to the methods has not been sufficiently developed. The methods of selecting the research sample were not sufficiently explained. Some questions arise: what was the subjective scope of the research - to whom exactly was it addressed, to employees of which sector, what positions, what organizations etc.; what was the spatial scope of the research - country, region, type/name of settlement center, etc. At the beginning, the reader was not informed directly about the country where the research was conducted and about the type of economic activity researched and its type (the restaurant chain is mentioned in point 5 of the paper). It seems that some information from point 6 of the article would be worth presenting within the specifics of the area covered by the study.
  4. It is necessary to know whether the sample selection for the research was random or purposeful and how exactly it was carried out. Does the obtained sample meet the criterion of representativeness? If so, about what group. It is worth explaining why the sample size was 335 employees. This has an impact on the possibility of interpreting the obtained research results.
  5. In section 3 (on methods), the methods used for data analysis were not properly listed and described (with the purpose of their application). References to them appear in the section devoted to the results (No. 4).
  6. The description of the methods also requires an explanation as to why certain elements were eliminated from the job satisfaction test.
  7. Due to the complexity of measuring the studied phenomena, it might be worth devoting a separate point to the issue of measuring the studied variables in the theoretical part.
  8. In the context of the title of the work, the question arises as to how the implementation of the study in the conditions of the COVID 19 pandemic could affect the research results. Would the results of such a pre-pandemic study be different? To what extent? It seems that the title of the article does not portend its content very well. Perhaps maybe you should take into account subtitle (e.g.chain restaurants, country, based on example of…).
  9. In order to make the presentation of research results easier to read, I propose to present the values of statistics in tables. In section 4.4. the fragment concerning the research by Hayes (2017) requires further development. Why was this model used.
  10. In conclusions, it would be worth referring to the specifics of the industry under study.

Other comments:

  1. The abstract requires some changes. it should be shortened. In addition, the purpose, material, all methods used and main conclusions should be clearly stated. Abstract also requires specifying the scope of the research - spatial and sectoral.
  2. The consistency of the text between paragraphs should be verified. I notice a certain inconsistency between the paragraph that begins with the words "Chinas's catering industry ...." and the previous one.
  3. It should be eliminated informal, imprecise terms such as "dark times", “even now”, “common wisdom” from the article.
  4. The title of item 2.2. it is too narrow for its content. The concept of "well-being" needs to be expanded.
  5. In point 3 of the work, the term "emotional disorder" appears, which has not been adequately explained.

Thank you

Author Response

Point 1: The difficulty in reading and evaluating the article is the fact that the tables and figures referenced in the text are not included. Perhaps the correct version of the text was not submitted for review. The above problem is very evident in the methods and results section of the article.

Response 1: I’m sorry for the inconvenience. We have inserted our tables and figures in the manuscript document.

Please see the revised version.  The figure1 and figure 2 were on Page 5 and 25. The table1 was on Page 12 and table 2-5 were on Pages 15-22. (in red).

Point 2: A lot of detailed hypotheses were formulated in the article. It would be easier for the reader to visualize them graphically. The missing figure 1 certainly contains valuable information on the measurement model. It seems that not all the hypotheses have sufficiently strong roots in the literature.

Response 2: We have inserted our tables and figures throughout our manuscript. We also revised our literature review to provide stronger evidence for these hypotheses.  

Please see the revised version on Page 4 and 9.  The figure1 and figure 2 were on Page 5 and 25.  (in red).

Point 3a: I believe that the part of work devoted to the methods has not been sufficiently developed. The methods of selecting the research sample were not sufficiently explained. Some questions arise: what was the subjective scope of the research - to whom exactly was it addressed, to employees of which sector, what positions, what organizations etc.; what was the spatial scope of the research - country, region, type/name of settlement center, etc.

Response 3a:      We have revised our methods section to clarify on these questions.

Please see the revisions on Pages 9-10. (in red).

Point 3b:At the beginning, the reader was not informed directly about the country where the research was conducted and about the type of economic activity researched and its type (the restaurant chain is mentioned in point 5 of the paper). It seems that some information from point 6 of the article would be worth presenting within the specifics of the area covered by the study.

Response 3b:      Thank you. We have revised our abstract to address these concerns.

Please see the revisions on Page 1. (in red).

Point 4: It is necessary to know whether the sample selection for the research was random or purposeful and how exactly it was carried out. Does the obtained sample meet the criterion of representativeness? If so, about what group. It is worth explaining why the sample size was 335 employees. This has an impact on the possibility of interpreting the obtained research results.

Response 4: Thank you for the question. We have re-written our Data Collection section to clarify on these questions.    

Please see the revisions on Page 10. (in red).

Point 5: In section 3 (on methods), the methods used for data analysis were not properly listed and described (with the purpose of their application). References to them appear in the section devoted to the results (No. 4).

Response 5: We have revised our Data Analysis section.    

Please see the revisions on Page 10. (in red).

Point ï¼–: The description of the methods also requires an explanation as to why certain elements were eliminated from the job satisfaction test.

Response 6: We have addressed the question in our Data Collection section.     

Please see the explanation on Page 10. (in red).

Point 7: Due to the complexity of measuring the studied phenomena, it might be worth devoting a separate point to the issue of measuring the studied variables in the theoretical part.

Response 7:  We have added a paragraph to emphasize their theoretical associations in the methods section.      

Please see the revisions on Pages 9-10. (in red).

Point 8:In the context of the title of the work, the question arises as to how the implementation of the study in the conditions of the COVID 19 pandemic could affect the research results. Would the results of such a pre-pandemic study be different? To what extent? It seems that the title of the article does not portend its content very well. Perhaps maybe you should take into account subtitle (e.g.chain restaurants, country, based on example of…).

Response 8: We have revised our results section to be clearer. Since our data were collected during the pandemic, our results were discussed in the context of the COVID pandemic, but were not directly related to COVID. We also deleted some unrelated references to COVID.

Please see the revisions on Page 28. (in red).

Point 9: In order to make the presentation of research results easier to read, I propose to present the values of statistics in tables. In section 4.4. the fragment concerning the research by Hayes (2017) requires further development. Why was this model used.

Response 9: Thank you very much. We explained the Bootstrap method on Pages 24-25.  

Please see the explanation on Pages 24-25. (in red).

Point 10: In conclusions, it would be worth referring to the specifics of the industry under study.

Response 10: We have added a paragraph of industrial implications.      

Please see the addition on Page 28. (in red).

Point 11: The abstract requires some changes. it should be shortened. In addition, the purpose, material, all methods used and main conclusions should be clearly stated. Abstract also requires specifying the scope of the research - spatial and sectoral.

Response 11: We have revised our abstract.    

Please see the revisions on Page 1. (in red).

Point 12: The consistency of the text between paragraphs should be verified. I notice a certain inconsistency between the paragraph that begins with the words "Chinas's catering industry ...." and the previous one.

Response 12: We have fixed these wording to be consistent.     

Please see the revisions on the whole paper. (in red).

Point 13: It should be eliminated informal, imprecise terms such as "dark times", “even now”, “common wisdom” from the article.

Response 13: We have changed these wording.     

Please see the revisions on Pages 2-3. (in red).

Point 14: The title of item 2.2. it is too narrow for its content. The concept of "well-being" needs to be expanded.

Response 14: We explained more about the concept of “well-being”.      

Please see the revisions on Page 4. (in red).

Point 15: In point 3 of the work, the term "emotional disorder" appears, which has not been adequately explained.

Response 15: We have deleted the unrelated word.      

Round 2

Reviewer 2 Report

Dear Authors. Thank you for your comments and responses.